# Genome-Wide Identification and Analysis of the NF-Y Transcription Factor Family in *Medicago sativa* L.

Tingting Song [1] , Jiawei Li [1] , Yuying Yuan [1] , Jinqiu Yu [1] , Yuqi Cao [1] , Hua Cai [2,] * and Guowen Cui [1,] *

1 College of Animal Science and Technology, Northeast Agricultural University, Harbin 150030, China; songtingting147@126.com (T.S.); lijiawei011@126.com (J.L.); yuanyuying001@163.com (Y.Y.); yjq0726@163.com (J.Y.); caoyuqi18166280980@163.com (Y.C.)
2 College of Life Science, Northeast Agricultural University, Harbin 150030, China
* Correspondence: caihuaneau@gmail.com (H.C.); cgw603@163.com (G.C.)

**Abstract:** The nuclear factor Y (NF-Y) gene family is an important transcription factor family consisting of three subfamilies, NF-YA, NF-YB and NF-YC, which are widely involved in plant growth and development, stress responses and other processes. In this study, we identified 64 members of the *NF-Y* gene family in the *M. sativa* L. (Xinjiang Daye) genome, including 11 *MsNF-YAs*, 33 *MsNF-YBs* and 20 *MsNF-YCs*. Analysis of conserved motifs indicated that each unit included unique compounds of motifs, although certain members lost some motifs. Conserved functional domain analysis showed that each subunit contained a specific set of functional domains. Analysis of cis-acting elements in the promoter region of the *MsNF-Y* genes identified a series of cis-acting elements associated with stress responses. In addition, the transcriptome data and qRT-PCR analysis showed that *MsNF-Y* genes were significantly induced or downregulated by alkali treatment. The results of this study may help to establish a basis for further cloning and functional studies of *NF-Y* genes in *Medicago sativa* and other related legume species.

**Keywords:** *Medicago sativa* L.; *NF-Y*; alkali stress; expression profiles





## 1. Introduction

Transcription factors (TFs) can specifically bind to cis-acting elements in the promoter regions of eukaryotic genes, thereby activating or inhibiting the transcription and expression of downstream genes at specific growth and developmental stages or in specific tissues. TFs have become a popular research topic in the field of plant abiotic stresses [1]. Nuclear factor Y (NF-Y), also famous as heme activator protein (HAP) or CCAAT-binding factor (CBF), is a complicated, heterotrimeric transcription factor that is universal in eukaryotes. In all plants, there are three unique and conserved subunits named NF-YA or HAP2, NF-YB or HAP3/CBF-A and NF-YC or HAP5/CBF-C in NF-Y [2]. First, the NF-YB and NF-YC subunits form a dimer in the cytoplasm, then they combine with NF-YA subunits in the nucleus to form a heterotrimer [3] and, finally, they bind to the CCAAT-box to modulate the transcription of specific genes [4]. In mammals and yeast, each NF-Y subunit is encoded by a single gene, while in plants, each NF-Y subunit has evolved into a multigene family. In *Arabidopsis thaliana*, there are 10, 13 and 13 *AtNF-YAs*, *AtNF-YBs* and *AtNF-YCs* [5], respectively, and 11 *OsNF-YAs*, 11 *OsNF-YBs* and 12 *OsNF-YCs* were identified in *Oryza sativa* [6]. In *Brachypodium distachyon*, 36 NF-Ys (7 *BdNF-YAs*, 17 *BdNF-YBs* and 12 *BdNF-YCs*) were characterized [7], while in *Brassica napus*, there were 33 (14 *BnNF-YAs*, 14 *BnNF-YBs* and 5 *BnNF-YCs*) [8] and in *Glycine max*, there were 68 (21 *GmNF-YAs*, 32 *GmNF-YBs* and 15 *GmNF-YCs*) [9].

In plants, NF-Y subfamilies are indispensable for a wide series of developmental processes (embryogenesis, flowering time, chloroplast biogenesis, seed germination) as well as tolerance to abiotic stresses [10]. *AtNF-YA1* is related to post-germinative growth arrest under salt stress [11]. Overexpression of *AtNF-YA5* results in enhanced drought

tolerance through activation of stress-responsive genes [12]. AtHAP5A binds to the CCAAT motif of *AtXTH21* to modulate freezing stress resistance in *A. thaliana* [13]. Ni et al. reported that miR169 and its downstream regulating gene, GmNF-YA3, are positive regulators of plant tolerance to drought stress in *A. thaliana* [14]. Overexpression of the *OsHAP2E* gene enhances photosynthesis and tiller (stems produced in grass plants) quantity and boosts both drought and salt tolerance [15]. Overexpression of *CdtNF-YC1* in hybrid bermudagrass results in raised drought and salt tolerance in transgenic seashore paspalum associated with the induction of a series of stress-responsive genes [16]. It is also worth mentioning that the *NF-Y* family genes are not redundant but have specific functions. The NF-Y family perform essential functions in plant development and plasticity, including the formation of lateral root organs, such as lateral roots and symbiotic nodules. In *M. truncatula*, *MtNF-YA1* and *MtNF-YA2* play an important role not only in the early stages of rhizobial infection but also in the later stages of RNS to mediate nodule organogenesis and the persistence of nodule hyphal tissue [17–19], and the *MtNF-YC1* and *MtNF-YC2* genes are required for nodule organogenesis [20,21]. Meanwhile, two NF-Y subunits of *Lotus japonicus*, *LjNF-YA1* and *LjNF-YB1*, were identified as direct transcriptional targets of the master symbiotic regulator Nodule Inception (NIN) as well [21]. Roderick W. et al. found that AtNF-YC-3, -4 and -9 can interact with bZIP proteins that are known to control ABA signaling [22]. NF-Y subunits show characteristic phenotypes associated with the disruption of ABA signaling and can have opposing roles in ABA signaling. In addition, *AtNF-YC2* and *NF-YB3* are upregulated after ER treatment and transfer from the cytoplasm to the nucleus, where they combine with AtbZIP28 to generate a transcriptional complex that upregulates the expression of ER stress-induced genes.

　　　Approximately more than half of cultivated land in the world is threatened by varying degrees of salinization [23]. According to previous reports, alkali stress can cause the deterioration of soil physical and chemical properties, contributing not only to low water potential [24] and ion toxicity [25] but also high pH stress [26]. These effects can inhibit the normal physiological metabolic process of plant cells, resulting in slowed growth and even withering and death in severe cases [26]. However, by a series of gene expression and product interactions, plants have evolved to be able to adjust to abiotic stresses at cellular and molecular levels. These interactions may help to restore ion balance by regulating intracellular pH [27]. Plants are also able to repair damage due to alkali stress and improve the rhizosphere environment so as to make it possible to develop and utilize saline–alkali soil which cannot be cultivated [28]. Cultivated alfalfa (*Medicago sativa* L.), a perennial autotetraploid (2n = 4x = 32) and cross-pollinated forage legume, is the most significant cultivated forage plant all over the world [29] because of its wide ecological adaptability, high nutritional value and ability to prevent soil salinization and desertification [30]. However, alkali salinity and drought combined with low temperature limit the production and yield stability of alfalfa. Enhancing the final yields and productivity of alfalfa is especially challenging for researchers due to the unpredictable nature of abiotic stress conditions during the growing season and a complex abiotic stress biology [31]. In many plants, an increasing number of studies have improved the wealth of information about the functions of *NF-Y* genes. Thanks to the publication of chromosome-level genome sequences based on XinJiang Daye [29] and Zhongmu No. 1 [32], the NF-Y family can be comprehensively analyzed in alfalfa now. In this study, through phylogenetic analysis, structural diversification and gene expression analysis under alkali stress, the *NF-Y* genes in *M. sativa* were systematically identified and characterized. The results of this study may help broaden our understanding of the functions of the *NF-Y* gene family in *M. sativa* and other related legume forages. These findings may help to identify candidate genes that can improve alkaline or other tolerance through genetic modification.

## 2. Materials and Methods

### 2.1. Plant Materials and Treatment

*Medicago sativa* (Xinjiang Daye) was used in this study. Seeds were planted in plastic cups filled with vermiculite, germinated and irrigated with 1/10 Hoagland nutrient solution once every two days. The plants were maintained in a greenhouse for 4 weeks (30 °C/25 °C day/night temperatures, 16/8 h of light/dark), after which the plants were treated with 150 mmol/L $NaHCO_3$. The treatment was continued for 48 h, and leaf tissues were collected at 0 h (CK), 3 h, 6 h, 12 h, and 48 h after the administration of an alkali treatment. Three plants from each time point served as the sample sources for RNA extraction and qRT-PCR analysis. All the fresh tissues were harvested, immediately frozen in liquid nitrogen and stored at −80 °C until extraction.

### 2.2. Identification and Characterization of the NF-Y Family in Alfalfa

*NF-Y* gene and protein sequences of *A. thaliana* were downloaded from the NCBI (National Biotechnology Information Center, https://www.ncbi.nlm.nih.gov, accessed on 9 August 2021) database, and the *M. sativa* genome was downloaded from Chen et al. [29]. The hidden Markov model (HMM) profiles of the NF-Y domains (PF02045, PF00808) were retrieved from the Pfam (http://pfam.xfam.org, accessed on 9 August 2021) database. A total of 162 genes in the alfalfa genome were identified as possible members of the MsNF-Y gene family using BlastP. Of these, 98 duplicate sequences were considered as alleles of 64 *MsNF-Ys* by McScanX [33]. Finally, the nucleotide and deduced amino acid sequences of *MsNF-Y* genes were confirmed for further analysis. The amino acid sequence lengths, molecular weights (MW) and pI values of all predicted *MsNF-Ys* were then determined by ExPASy (https://www.expasy.org, accessed on 9 August 2021).

### 2.3. Phylogenetic and Multiple Sequence Alignment Analysis of NF-Y Proteins

To study the phylogenetic relationships of the NF-Y family, the full-length amino acid sequences of NF-Y members of alfalfa and *Arabidopsis* were used to construct phylogenetic trees by the neighbor joining (NJ) method with 1000 bootstrap replicates through MEGA 6.06 [34] software. Multiple sequence alignments of the 3 NF-Y subunits in alfalfa were aligned by DNAMAN (Lynnon Biosoft) 6.0.

### 2.4. Conserved Motifs, Conserved Domains and Cis-Element Analysis of the NF-Y Gene Family in Alfalfa

Conserved motifs and domains of *NF-Y* genes were predicted by MEME (https://meme-suite.org, accessed on 12 August 2021) and the Batch CD-search (https://www.ncbi.nlm.nih.gov/Structure/bwrpsb/bwrpsb.cgi, accessed on 12 August 2021) online program. MEME was optimized with the following parameters: the optimal width of each motif was between 6 and 50 and the maximum number of motifs found was 10. The Plant-CARE database (http://bioinformatics.psb.ugent.be/webtools/plantcare/html, accessed on 12 August 2021) was used to predict and analyze the cis-elements of 2000 bp up-stream sequences of each *MsNF-Y* gene.

### 2.5. Transcriptome Sequencing and Quantitative Real-Time PCR (qRT-PCR)

Total RNA was extracted from each sample using an UItrapure RNA kit (CoWin Biotech, Beijing, China) and a NEBNext Ultra™ RNA Library Prep Kit(NEB, Ipswich, MA, USA) for Illumina library construction. All the libraries were sequenced using the Illumina (San Diego, CA, USA) HiSeq X-ten platform with PE150. The paired-end reads that were retained after quality control were mapped to the genome using HISAT2 (v.2.0.4) [35] with default parameters. Fragments per kilobase of exon per million mapped reads (FPKM) was used to analyze the expression levels of *MsNF-Y* genes by TopHat and Cufflinks [36]. The FPKM values for *NF-Ys* genes were utilized to generate a heatmap and rows were normalized using TBtools software [37].

First-strand cDNA was synthesized from 1 μg of total RNA using a HiScript III 1st Strand cDNA Synthesis Kit (Vazyme Biotech, Nanjing, China), according to the manufacturer's instructions. Gene-specific primers were designed using Primer Premier [38] software. In each reaction, the GAPDH (accession no.: XM_003601780.1) gene was used as an internal reference gene. The relative gene expression levels were calculated according to the $2^{-\Delta\Delta C(t)}$ method. SPSS 25 (IBM) was used for data analysis, and the Duncan method was used to compare the significant differences between treatments ($p < 0.05$), which were marked with different lowercase letters. All data are the means ± standard error of 3 biological replicates with 3 technical replicates. All the primers used for qRT-PCR are listed in File S1.

## 3. Results

### 3.1. Identification of NF-Y Family Genes in Medicago sativa L.

A total of 162 *NF-Y* genes in *Medicago sativa* L. Xinjiang Daye were identified, including 30 *NF-YA*, 89 *NF-YB* and 43 *NF-YC* genes. By circos, they are presented as many sets of alleles (Figure S1). After picking the longest in every set, each *MsNF-Y* gene was named according to its order of distribution on the eight chromosomes (File S2). For the 64 putative MsNF-Y proteins, their molecular weights (Mws) and isoelectric points (pIs) were also determined using the ExPASy online service (File S3). The protein sequences encoded by the *MsNF-Y* genes ranged in length from 99 amino acids for MsNF-YC13 to 379 amino acids for MsNF-YC9, with an average of approximately 220 amino acids. The predicted Mws of the MsNF-Y proteins ranged from 11.15 kDa to 41.59 kDa and the theoretical pI values ranged from 4.18 to 9.91. The results confirmed that 64 MsNF-Y proteins had large differences in sequence and protein characteristics. Moreover, the number of NF-Y gene family members in *M. sativa*. L was greater than those in *M. truncatula* and *A. thaliana*. Thus, the genome of autotetraploid alfalfa was more complex.

### 3.2. Phylogenetic Analysis of MsNF-Y Genes

MEGA 6.06 software was used to construct a phylogenetic tree of 64 *M. sativa* and 36 *A. thaliana* NF-Ys (Figure 1). According to the phylogenetic tree, 11 MsNF-YAs and 10 AtNF-YAs were assigned to the NF-YA group. Thirty-three MsNF-YBs and 10 AtNF-YBs were classified in the NF-YB group, of which 2 MsNF-YBs (9 and 28), AtNF-YB9 (LEC1, LEAFY COTYLEDON 1) and AtNF-YB6 (L1L, LEC1-LIKE) [39], were assigned to the LEC subgroup (Figure 1); 20 MsNF-YCs and 13 AtNF-YCs were classified into the NF-YC group; while 3 AtNF-YCs (10, 12 and 13) had no homologous genes in *M. sativa*. There were 6 MsNF-YC members similar to AtNF-YC11 (NC2α, negative cofactor 2α) [40]. Therefore, according to the similarity with AtNC2α, the NF-YC family was divided into two subgroups, named the NC2α and non-NC2α subgroups (Figure 1).

### 3.3. Analysis of Conserved Motifs, Domains and Multiple Alignments

A conserved motifs analysis was performed to support a phylogenetic reconstruction by transferring 64 NF-Y protein sequences to the online MEME web server (Figure 2B). Ten conserved motifs were identified in the three subunits of NF-Y proteins (Figure S2) and the results were mostly consistent with those of the phylogenetic analysis (Figure 2A). Motif 7-5 was detected in almost all of the NF-YA proteins except MsNF-YA6. The NF-YB proteins shared a similar motif distribution containing motif 3-1-2. Non-MsNC2α members in MsNF-YC shared motif 4-1-6 (except for MsNF-YC16 and 13, which, respectively, lost motifs 4 and 4-6), while motif 7-6 was detected in all six MsNC2α subgroup members.

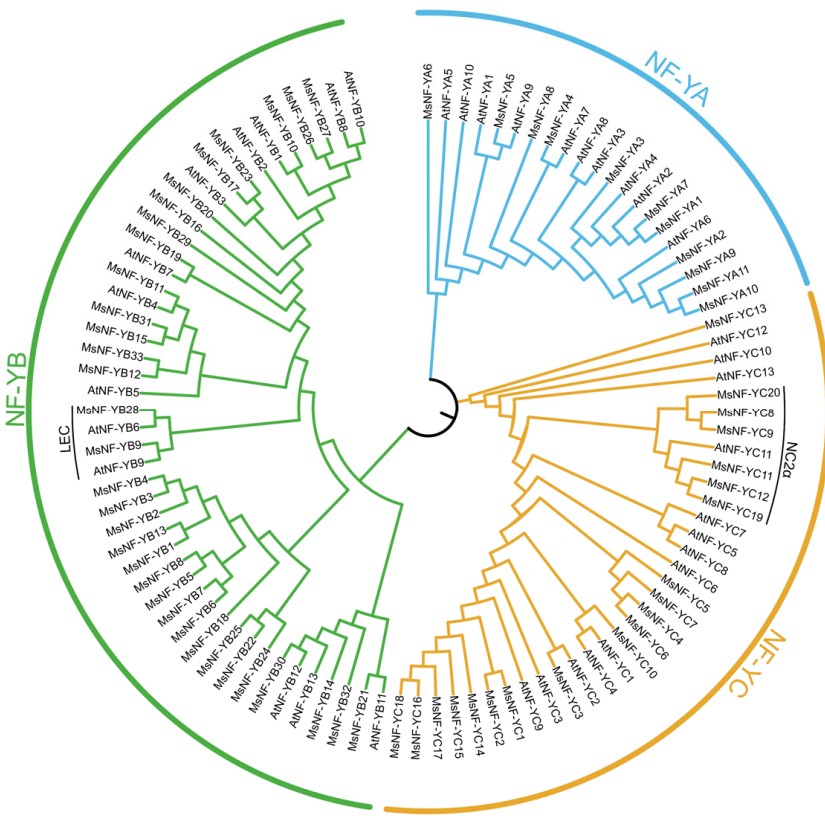

**Figure 1.** Phylogenetic tree of *NF-YA*, *NF-YB* and *NF-YC* sequences from *Medicago sativa* and *Arabidopsis thaliana*. The blue, green and yellow lines and branches respectively represent the NF-YA, NF-YB and NF-YC families.

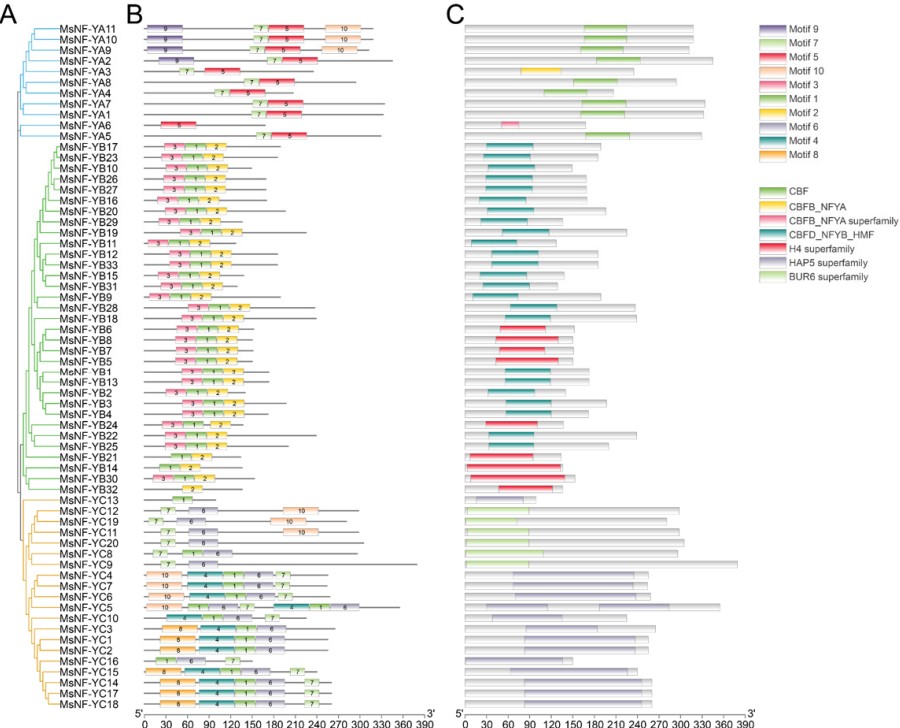

**Figure 2.** (**A**) Phylogenetic relationships, (**B**) conserved protein motifs and (**C**) conserved domains of MsNF-Y. The boxes with different colors represent the conserved motifs listed in Figure S2 and the conserved domains.

The distribution of conserved domains was determined by the conserved domain database of NCBI (Figure 2C). MsNF-YAs contain conserved CBF (CCAAT-binding transcription factor) and CBFB_NFYA (CCAAT-binding transcription factor-B_NFYA) domains or the CBFB_NFYA superfamily. Through multiple sequence alignment and functional domain analysis, MsNF-YA members were found to contain an NF-YB/C-interaction domain and a highly conserved DNA-binding domain (Figure 3A). MsNF-YBs contain conserved domains in the H4 superfamily or CBFD_NFYB_HMF (CCAAT-binding transcription factor-D_NFYB_histone-like transcription factor) (Figure 2C). Similarly, from multiple sequence alignments, the MsNF-YB member contains two NF-YC-interaction domains separated by seven amino acids, one NF-YA-interaction domain and one DNA-binding domain (Figure 3B). The MsNC2$\alpha$ subgroup contains BUR6 (bypass UAS requirement 6) superfamily conserved domains, while the non-MsNC2$\alpha$ subgroup contains HAP5 (histone or heme-associated protein 5) superfamily conserved structures (Figure 2C). It is worth noting that the non-MsNC2$\alpha$ of MsNF-YC members contains two NF-YA-interaction domains, one NF-YB-interaction domain and one two-amino acid-long DNA-binding domain (A and R amino acid residues), which are highly conserved among the non-MsNC2$\alpha$ subgroup members of MsNF-YC. However, in the MsNC$\alpha$ subgroup, the NF-YA- and NF-YB-interaction domains were mutated, though the DNA-binding domain was still conserved (Figure 3D). These 11 MsNC2$\alpha$ members were only 53.09% similar to AtNF-YC11. It can be seen that the three subunits have different functional domains and have different functions before forming the heterotrimer. As a transcription factor complex, all three subunits contained unique DNA-binding domains, which showed that the functions regulated by the NF-Y transcription factor would be more numerous.

*3.4. Promoter Region Analysis of MsNF-Y Genes*

The characterization of cis-regulatory elements in the promoter region can inform the tissue-specific or stress-response expression patterns of genes [41]. The 2000 bp promoter region located upstream of the translational start site (ATG) of each *MsNF-Y* gene was analyzed using the online PlantCARE software to determine its feasible regulatory mechanism. It was discovered that several cis-regulatory elements related to abiotic stress were found in the promoter region of the *MsNF-Y* gene. (Figure 4). Thirteen cis-elements were observed to be stress-responsive elements, including TC-rich repeats, DRE, MYC, MYB, LTR, MBS, CCAAT-box, AT-rich element, ARE, GC-motif, as-1, WUN-motif, W box, which were involved in defense and stress responsiveness, low-temperature stress, and drought inducibility, especially the 11 MYCs in *MsNF-YA6* and *-YB16* and 11 MYBs in *MsNF-YC4*. It should be pointed out that in the three subunit groups, most genes contained several stress-responsive MYB and MYC cis-elements. These results suggested that the *MsNF-Y* genes might be involved in responses to multiple abiotic stresses.

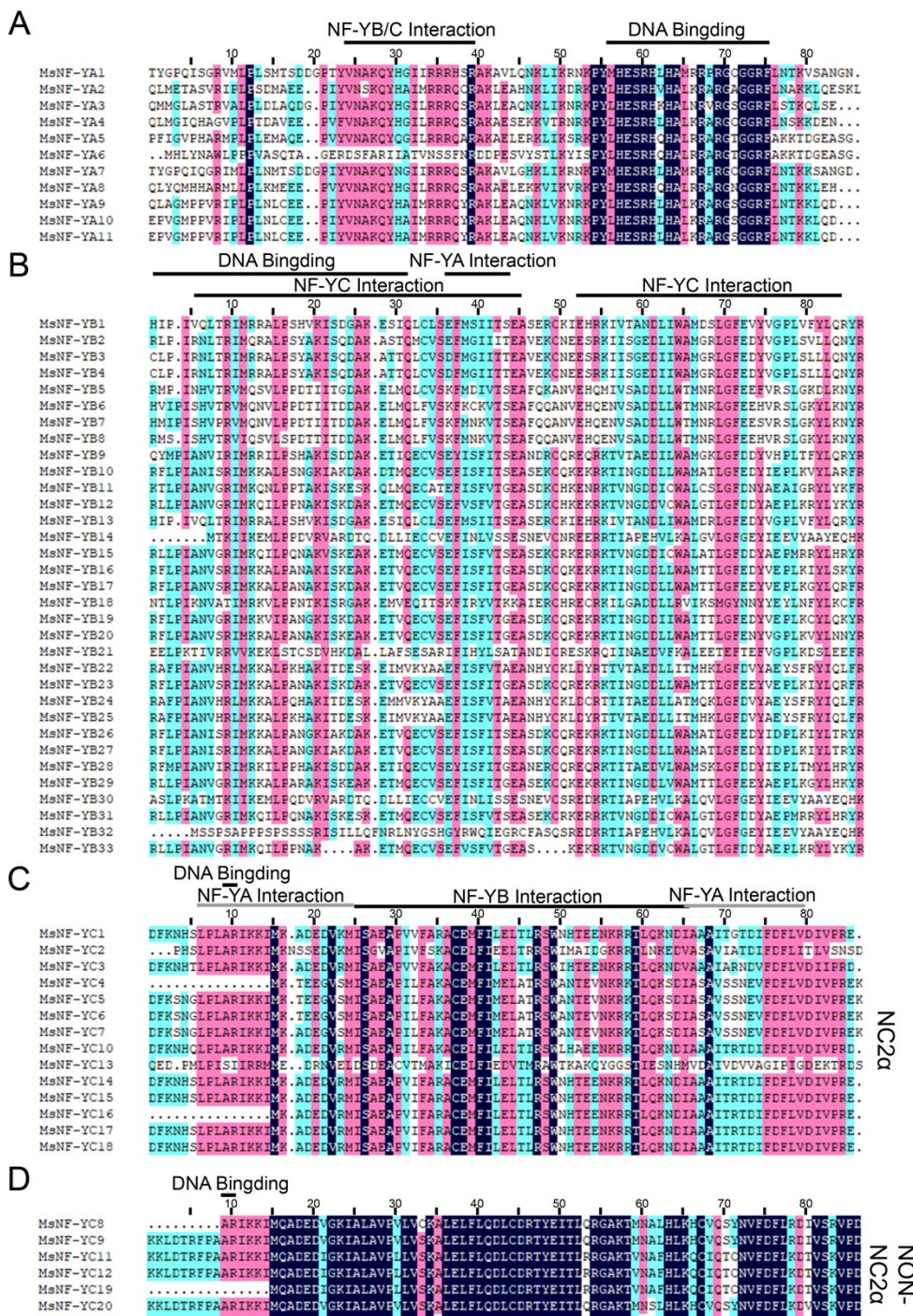

**Figure 3.** The multiple sequence alignments of (**A**) MsNF-YA, (**B**) MsNF-YB, (**C**) non-MsNC2α and (**D**) MsNC2α. Black and grey lines show the interaction and binding domains of MsNF-Y proteins.

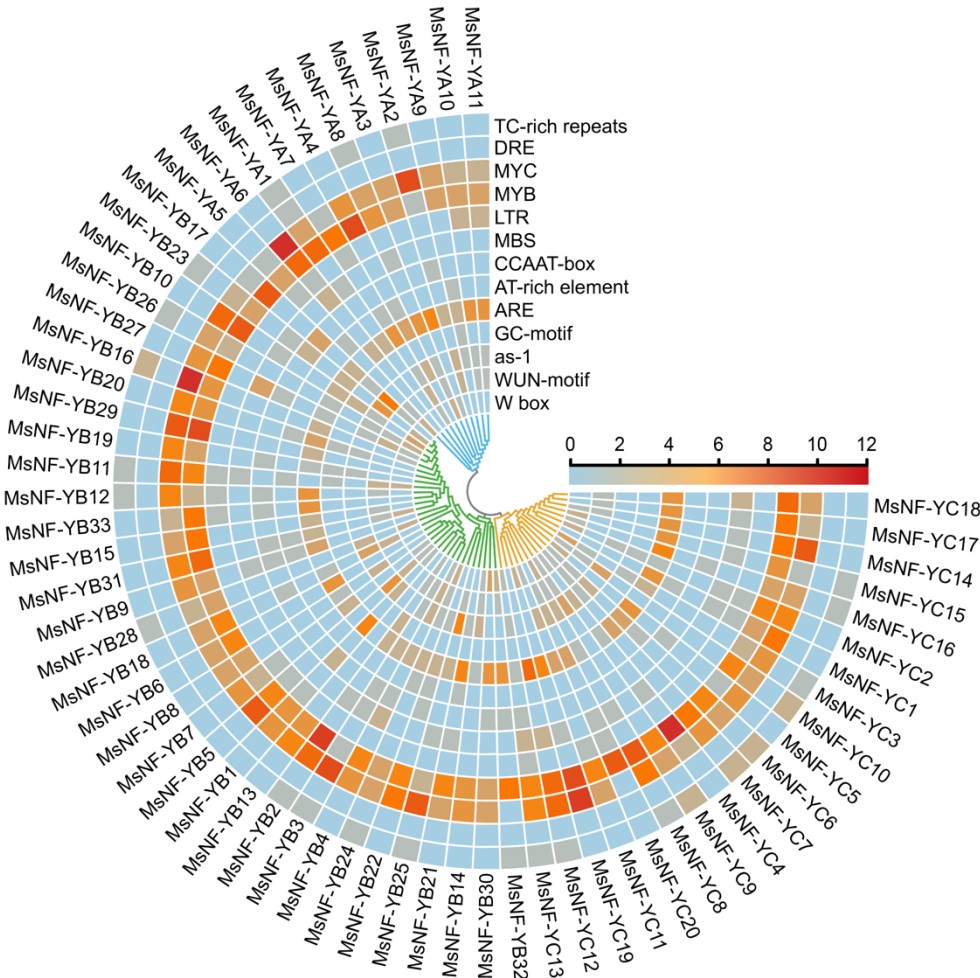

**Figure 4.** Cis-acting elements analysis of the promoter region of *MsNF-Y* genes. The cis-elements located in the ~2000 bp promoter sequence (upstream of the start codon) of all *NF-Y*s were analyzed via the PlantCare website. The color indicates the quantity of cis-acting elements (details shown in File S4).

### 3.5. Transcriptome Data and Expression Patterns of MsNF-Y Genes under Alkali Treatment

RNA-seq data for alkali stress were retrieved to investigate the potential response of *MsNF-Y* genes to abiotic stress (Figure 5A). Under alkali stress, 24 out of 64 *MsNF-Y* genes were identified as non-expressed genes which all gather in *NF-YB* and *NF-YC*. Only 10 of 40 detected genes (*MsNF-YA2, 3, 4, 5, 9; -YB27, 28, 29, 32*; and *-YC5*) showed varying degrees of downregulation, which means that most *MsNF-YB* and *-YC* genes can be positively regulated by alkali treatment in different ways.

In this study, we randomly checked the expression patterns of eight *MsNF-Y* genes that contain more stress-related elements in response to alkali stress (150 mmol/L NaHCO3) through qRT-PCR analysis (Figure 5B). Overall, the eight *MsNF-Y* genes all responded to alkali stress treatments. There were different expression patterns among the genes of three subunit group members. After alkali treatment, expression of the *MsNF-YA1, -YB10, -YB25* and *-YC10* genes increased in the early stage and then decreased significantly. However, the peak values of four genes differed. Those of *MsNF-YA1* and *-YC10* appeared at 12 h after treatment, while that of *MsNF-YB10* appeared at 6 h and that of *MsNF-YB25* at 3 and 12 h after treatment. The expression levels of *MsNF-YA2* and *-YC11* increased with fluctuations, while that of *-YC20* increased continuously. By contrast, the expression level of *MsNF-YA8* showed a remarkable decline, although it recovered to control level 48 h after treatment. Seven out of eight detected genes showed upregulation and reached a relatively high level, especially *MsNF-YC11*, whose expression level was 16.5 times that of the untreated group.

These results were similar to the RNA-seq data and indicate that all 40 detected genes could be involved in the response to alkali stress with different functions.

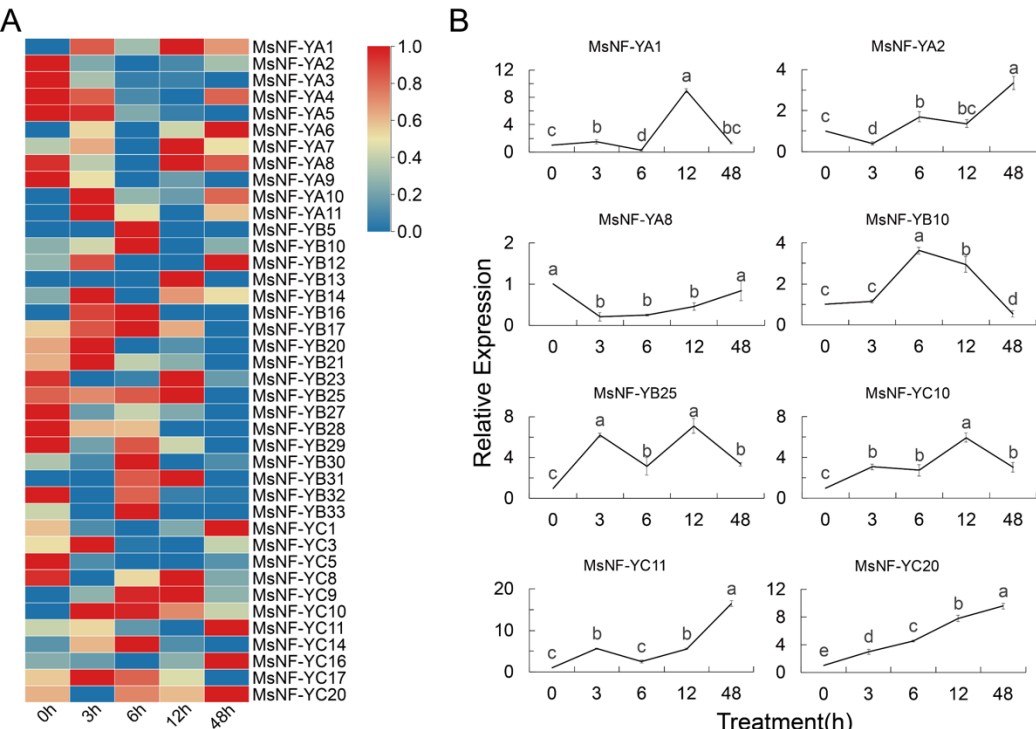

**Figure 5.** Heatmap of (**A**) RNA-seq data for the MsNF-Y gene family and expression profiles of 8 *MsNF-Y* genes. (**B**) Under alkali treatment by qPCR: alfalfa seeding in response to 150 mM NaHCO₃. Data are means ± SE of three replicates. Different lowercase letters indicate significant differences ($p < 0.05$).

## 4. Discussion

Unlike yeast and mammals, plants have a large NF-Y family, commonly with multi-gene families encoding each subunit. The NF-Y gene family has been identified in many plants, including Arabidopsis [5], rice [6], maize [42], barley [43], wheat [44] and rape [8]. Other studies have shown that members of the NF-Y gene family exhibit different expression patterns in different species, and there are redundancies and functional differences in their regulation of plant growth and development [45]. In this study, 64 *MsNF-Y* genes were identified in *M. sativa* L. Xinjiang Daye. Compared with 36 *AtNF-Y* genes in diploid Arabidopsis, 33 *BnNF-Y* genes in allotetraploid rape and 35 *TaNF-Y* genes in allohexaploid wheat, the number of *MsNF-Y* genes is much higher than those of the above species. One possible reason is that alfalfa is an autotetraploid and its genome is highly heterozygous and repetitive [29]. The heterotrimeric nature of the NF-Y family in plants indicates there might be a huge number of NF-Y complexes formed. The result would be the formation of a flexible, combinatorial transcription factor system that may allow for subtle adaptations to numerous environmental conditions [46].

To investigate the developmental and evolutionary relationship between *M. sativa* and *A. thaliana*, a phylogenetic tree was constructed using all the NF-Y protein sequences of *M. sativa* L Xinjiang Daye and *A. thaliana* (Figure 1). The branches on this phylogenetic tree were quite similar to those that can be found in previous reports on Arabidopsis [5], which implied that most NF-Y proteins are homologous. It is worth noting that AtNF-YC10 (AtDpb3-1), AtNF-YC12, AtNF-YC13 (AtDpb3-2), and AtNF-YC5, 6, 7 and 8 were divided separately and it seemed that there was no corresponding homolog in *M. sativa*. However, there were six *NF-YC* genes that were highly homologous to *AtNC2α* (*AtNF-YC11*) in *M. sativa*. There are *NC2* and *Dpb3/Dpb4* genes that are highly conserved, from yeast to

mammals. The *AtNC2* and *AtDpb4* genes are well known to be replicated, while the *At-Dpb3* gene is not expanded in plants [47]. Therefore, NF-Y genes may be more important for gene regulation processes, leading to changes in plant developmental pathways.

Due to the structural similarity, NC2a might be included in the NF-Y gene family. However, this gene has been shown to be an outlier in the phylogenetic analysis of NF-Y proteins in *Arabidopsis*, wheat and *Brachypodium distachyon* [5,7,44]. NC2α does not functionally overlap with NF-Y; NC2α associates with TBP to bind TATA boxes in core eukaryotic promoters [40]. However, some researchers thought that all of the plant NF-YC11 homologs showed unique N-terminal sequences that made them distinctly different from other NF-YCs and identified them as a subgroup of the NF-YC group [48]. In this study, 64 members of the MsNY-F family with similar domains were screened through hidden Markov model (HMM) profiles of the NF-Y domain (PF02045, PF00808). Among them, the NF-YC group (20 genes) was divided into two subgroups, NC2α (6 genes) and non-NC2α (14 genes) (Figure 1). To determine whether the members of the NC2a subfamily in *M. sativa* belong to the NF-Y group, evidence beyond their conserved domains, such as heterotrimer formation during transcription, is required.

*AtNF-Y* also has an important role in plant response to abiotic stresses, such as drought, salt, cold and heat. *NF-Y*s have been identified as regulators of drought tolerance in different plant species and regulate drought stress responses independently of ABA regulation [45]. However, the regulation of the *NF-Y* gene under alkali stress has rarely been reported. Sequencing of the Xinjiang Daye alfalfa genome has been completed, which provides a good opportunity for the discovery of the function of the *NF-Y* genes in alfalfa. The specificity of *NF-Y* gene expression will help clarify the molecular mechanism of alfalfa. In the present study, we found that different subunits of the NF-Ys family exhibited different expression patterns in response to alkali stress. Most of the *NF-YA*s and *NF-YC*s had the highest expression at late stages of alkali stress (48 h), while *NF-YB*s responded to alkali stress and upregulated expression in the midterm (3–12 h). Several genes of the NF-Y family which contain more MYB or MYC elements in their upstream sequences were selected for qPCR analysis, and these elements were shown to be actively involved in the abiotic stress response of plants [49–51]. The results showed that *NF-Y*s were closely associated with alfalfa response to alkali stress, and most *MsNF-Y* genes were significantly up- or downregulated in response to alkali treatment. Therefore, we believe that *MsNF-Y*s might respond to abiotic stress through different pathways.

## 5. Conclusions

To summarize, this study identified and characterized 64 *NF-Y* genes in *M. sativa* L. Xinjiang Daye and explored their phylogenetic relationships. Through expression analysis, it was found that most *MsNF-Y*s respond to alkali stress. The diversity of NF-Y subunits and their many potential combinations, along with possible redundancies and differences in function, represent a major challenge for studies that aim to determine the functions of diverse subunits under abiotic stresses in alfalfa.

**Supplementary Materials:** The following supporting information can be downloaded at: https://www.mdpi.com/article/10.3390/agronomy12051237/s1, Figure S1: Synteny and allelic relationship of *MsNF-Y* family genes; Figure S2: Schematic diagram of motifs of MsNF-Y proteins; File S1: Primers for PCR amplification; File S2: Naming table of *MsNF-Y* family genes; File S3: Basic physicochemical properties of *MsNF-Y* family genes; File S4: Details of cis-elements of *MsNF-Y* family genes.

**Author Contributions:** T.S., H.C. and G.C. conceived and designed the project and the strategy; T.S. and J.L. performed the experiments; T.S., J.L., J.Y., Y.Y. and Y.C. contributed to the data analysis. All authors have read and agreed to the published version of the manuscript.

**Funding:** This research work was funded by the National Natural Science Foundation of China (31872998).

**Institutional Review Board Statement:** Not applicable.

**Informed Consent Statement:** Not applicable.

**Data Availability Statement:** Not applicable.

**Conflicts of Interest:** The authors declare no conflict of interest.

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
