# Peer review of "Genome-Wide Identification and Analysis of the NF-Y Transcription Factor Family in Medicago sativa L."

_agronomy, doi:10.3390/agronomy12051237_

Round 1
Reviewer 1 Report
See attachment.

Reviewer 2 Report
In this work Song and colleagues identify the NF-Y Transcription Factor family members in Medicago sativa. They study the phylogenetic relationships, the conservation of the domains and motifs, and the cis-element presence in their promoters. They also show gene expression data of the NF-Y members under alkali treatment. The authors focus their study on the role of this family in the response to abiotic stresses, and their findings are interesting as a foundation for further functional characterization.
In my opinion, the authors should mention the role of NF-Y during nodulation in the introduction. Several NF-Y family members are key regulators of nodule formation in Medicago and Lotus. The phylogeny would be more interesting if they include in the phylogeny more nodulating species, such as M. truncatula and lotus, and non-nodulating species such as Arabidopsis, highlining the genes functionally characterized as regulators of nodule formation.
The phylogenetic tree method is an important part of this research work. The method of building the tree should be expanded and justified.
How the RNA-seq libraries were built? How they were sequenced and how many reads were obtained?
Line 271: “A.” should go in italic.
Line 304. In the sentence “Gene expression analysis revealed that NF-Ys were closely related to and stress in alfalfa.” The “to and stress” seems wrong in the sentence.
Line 309: M. sativa in italic.
